# Differential RNA-Seq Analysis Predicts Genes Related to Terpene Tailoring in *Caryopteris × clandonensis*

**DOI:** 10.3390/plants12122305

**Published:** 2023-06-13

**Authors:** Manfred Ritz, Nadim Ahmad, Thomas Brueck, Norbert Mehlmer

**Affiliations:** Werner Siemens Chair of Synthetic Biotechnology, Department of Chemistry, Technical University of Munich (TUM), 85748 Garching, Germany; manfred.ritz@tum.de (M.R.); nadim.ahmad@tum.de (N.A.)

**Keywords:** terpene biosynthesis, cytochrome p450, *Caryopteris × clandonensis*, long read sequencing, transcriptomics, chemical diversity, volatile compound

## Abstract

Enzymatic terpene functionalization is an essential part of plant secondary metabolite diversity. Within this, multiple terpene-modifying enzymes are required to enable the chemical diversity of volatile compounds essential in plant communication and defense. This work sheds light on the differentially transcribed genes within *Caryopteris × clandonensis* that are capable of functionalizing cyclic terpene scaffolds, which are the product of terpene cyclase action. The available genomic reference was subjected to further improvements to provide a comprehensive basis, where the number of contigs was minimized. RNA-Seq data of six cultivars, Dark Knight, Grand Bleu, Good as Gold, Hint of Gold, Pink Perfection, and Sunny Blue, were mapped on the reference, and their distinct transcription profile investigated. Within this data resource, we detected interesting variations and additionally genes with high and low transcript abundancies in leaves of *Caryopteris × clandonensis* related to terpene functionalization. As previously described, different cultivars vary in their modification of monoterpenes, especially limonene, resulting in different limonene-derived molecules. This study focuses on predicting the cytochrome p450 enzymes underlying this varied transcription pattern between investigated samples. Thus, making them a reasonable explanation for terpenoid differences between these plants. Furthermore, these data provide the basis for functional assays and the verification of putative enzyme activities.

## 1. Introduction

*Caryopteris × clandonensis* is an ornamental plant, also known as “Bluebeard”, which is phylogenetically classified in the *Lamiaceae* family. It is easily cultivated and rich in volatile compounds. These, and other molecules detected and described, are terpenes, e.g., α-copaene, limonene, or δ-cadinene [1], terpene derivates, e.g., keto-glycosides, clandonosides, and harpagides [2], as well as the pyrano-juglon derivate α-caryopteron [3]. The species’ essential oil was found to display mosquito-repellent activity; however, the active agent for this mode of action was not yet detected [4]. The *Lamiaceae* family is known to harbor an interesting and valuable profile in secondary metabolites, including terpenoids, flavonoids, and phenylpropanoids [5,6,7]. These compounds play important roles in the plant’s interaction with its environment [8,9] as for the defense against abiotic and biotic stresses [10]. They also harbor potential in pharmaceutical or industrial applications, as seen for taxol [11], menthol [12], malvidin [13], isoliquiritigenin [14] or umbelliferone [15]. In general, terpenes and terpenoids are a molecule class, which is produced in vast varieties by flowering plants [16] and is involved in a wide range of biological activities. Essential oils and their monoterpenes, such as α-pinene and limonene, were investigated in terms of their anti-inflammatory and virucidal activity in recent studies [17,18,19]. Moreover, other terpenoids employ antibacterial properties [20] while others act as insecticides [4], are used as allelochemicals [21], or as attractants for pollinators [22]. The backbone of plant-derived terpenes is produced via the mevalonate pathway. For this, the precursors dimethylallyl diphosphate (DMAPP) and the functional isomer isopentenyl pyrophosphate (IPP) can be connected via isoprenyl diphosphate synthases (IDS) to form larger units of terpenes. IPP consists of five C-atoms (hemiterpene) whereas, through condensation of IPP and DMAPP via IDS monoterpenes (C10), sesquiterpenes (C15), diterpenes (C20), and higher terpene structures are built [23]. Further tailoring of these basic terpenes is conducted by terpene synthases (TPS) and cytochrome p450 enzymes (CYPs). Plant TPS mediate complex carbocation reactions, resulting in various cyclic structures of higher terpenes [24,25]. These can be divided into eight subfamilies (TPSa-h) which can be clade- or even species-specific [26]. The first step in tailoring monoterpenes is hydroxylation. Subsequently, CYPs are mediating a plethora of further reactions to enhance the functionalization (carboxylation, acetylation or forming peroxides) [27,28]. Due to their promiscuity towards substrates, only a few enzymes are necessary to yield various terpenoid structures and, therefore, differences in their functions and modality [29]. Multiple sequences of different source organisms are available in curated databases [30,31]. These allow easy access to the genetic information on these enzymes. With CYPs occurring in all living organisms [30], the enzymes, similarly to TPS, are divided for better identification, whereas specific CYP families are reserved for each type of organism. Plant CYP families can be found in CYP71-99 and CYP701-999, and in a four-digit scheme from CYP7001-9999 [32]. The categorization into these classes is dependent on sequence similarity. The same family (Arabic number) needs matching amino acids ≥40% and the subfamily (Arabic letter) ≥55% [33]. Therefore, the CYP76S40 [34] is the 40th individual enzyme from the CYP76S subfamily and the CYP76 family. This way, after annotation, contaminating sequences can be discarded solely due to their classification in a non-plant CYP family.

One approach to elucidate variations in the enzymatic makeup and investigate the sequences underlying terpene diversity is to compare differentially expressed gene (DEG) products at a quantitative level using modern bioinformatics tools. Differences in the metabolite profile exist during different stages of plant growth [35]. Different genes are regulated from seedlings to mature plants to translate their genomic information into proteins and interact in plant differentiation, protection or communication, depending on their developmental state [36]. During plant breeding, deletion, duplications, mutations or fragmentations can occur. Therefore, a distinct set of genes varies in its nucleotide code and their transcription or translation rate, resulting in different phenotypes in the mature plant [37]. The data can be levied and evaluated regarding efficacy to investigate these differences. The number of transcripts does not solely result in higher protein outcome, but also in, respectively, higher concentrations of secondary metabolites. Therefore, differential expression analysis can identify genes or gene products responsible for either the stress response mechanisms observed for abiotic stressors, such as drought or radiation, or as has been shown for biotic stressors, such as pests and plant reactions to herbivores [38]. Typical DEG experiments harness the up- and down-regulation of genes after induction or shock, e.g., during exposure to chemicals [39] or different environments [40]. Another possibility is the investigation of specific traits of plant cultivars due to their variations between hybrid plants [41]. Previously, the variations in *Caryopteris × clandonensis*’ volatile compound setup was investigated, and a difference in the synthesis of limonene-derived molecules (LDM) was observed [1]. The cultivar Dark Knight was detected to harbor a low amount, whereas Pink Perfection shows high amounts of LDM. These variations were discovered without a distinct change in their TPS or CYP makeup.

To that end, we show that the identification of terpene variety between different plant cultivars can be pursued on a molecular level using a quantitative bioinformatics method such as RNA-Seq analysis. Furthermore, we focus on terpene functionalization enzymes, especially cytochrome p450 enzymes, to elucidate the mechanisms behind the variations in monoterpene modifications as seen for limonene [1].

## 2. Results and Discussion

### 2.1. RNA Sequencing and Mapping Quality

Samples subjected to short-read sequencing were taken from leaves of six *Caryopteris × clandonensis* cultivars known to show differences in their LDM profile, Dark Knight (DK), Grand Bleu (GB), Good as Gold (GG), Hint of Gold (HG), Sunny Blue (SB), and Pink Perfection (PP). Sequencing was performed using an Illumina NovaSeq platform, which generated about 20 million raw reads in bases for each sample. The reads were processed to remove low-quality reads, bases, and adapter sequences, resulting in the clean reads used for downstream analysis. After this purification step, a loss of 5.0 to 14.9 million bases was seen between the samples. In Table 1, the run as well as cleaning and mapping statistics are summarized. The Q20 and Q30 scores indicate the sequencing quality, with Q30 indicating a lower error rate than Q20. This experiment’s high Q20 and Q30 scores suggest that the sequencing quality was highly sufficient, with only a few sequencing errors. Moreover, the clean reads exhibit a slight increase in quality scores, persistent throughout all samples.

The available genome sequences from *Caryopteris × clandonensis* PP [1] were subjected to further cleaning and improvement steps to curb the influence of contamination. A binning algorithm, MetaBAT2 [42], usually used for metagenomic data, was used on the long-read assembly of the genome and differentiated into 40 bins. The completeness and contiguity were checked and, in summary, the 782 scaffolds/848 contigs, which add up to 344 Mb with a genome completeness score of 96.8%, were reduced to 53 scaffolds/88 contigs, which add up to 298 Mb and a BUSCO score of 96.5%. The utilized BUSCO gene sets belonged to the closest affiliate *Eudicotidae*. Detailed information can be found in Appendix A. This refined genome was used as a reference for mapping the short-read sequences. A preliminary mapping of DK transcripts on the respective long-read genomic data, compared to mapping the transcripts on the PP genomic data, revealed an increased assignment of unique reads. Thus, the genome of *Caryopteris × clandonensis* PP was chosen as a mapping reference for both cultivars, DK and PP, resulting in a more comprehensive downstream analysis. The exact mapping counts for the different methods can be found in Appendix A.

The percentages of reads mapped to the reference genome, as seen in Table 1, indicate the data accuracy and low presence of contaminating DNA. The amount of uniquely mapped reads is also an important metric, as it indicates the proportion of reads that map to a unique location in the reference genome. A high percentage of uniquely mapped reads (greater than 70%) is desirable, reducing the possibility of mapping errors or ambiguous mapping locations [43]. In our setting, we were able to accurately map between 85.8% and 87.8% of the sequences, indicating that a large proportion of the reads were successfully located on the provided genome. Furthermore, the percentage of uniquely mapped reads ranged from 75.4% to 82.0%, which is reasonably high and suggests that the quality of the sequencing reads was sufficient to allow for exact mapping and is suited for downstream analysis. The observed duplication rates varied between 5.7% and 11.3%, and are well-known in plant transcript mapping due to transcript isoforms [44].

### 2.2. Identification of DEG

To identify the mechanism behind the modification of LDM, we wanted to focus on the DEGs between the cultivars of *Caryopteris × clandonensis*. Therefore, the mapping data were subset and pooled into highly LDM-positive (SB, PP) and highly LDM-negative (DK, GB) cultivars. The cultivars GG and HG were neither highly LDM-positive nor highly LDM-negative, therefore both were disregarded during the initial DEG analysis. From the 29,210 predicted genes in the mapping reference, 23,477 were observed to map in all investigated sets. The DEGs were filtered using a log2 fold-change cutoff of absolute values greater than 1, and an adjusted *p*-value of a minimum of 0.05, thereby the values for each cultivar were transcribed at least two-fold. The values fitting these parameters are highlighted in green; those which were disregarded during further analysis, because of not fitting the parameters, are shown in red. Compared to the genes close to the middle, there are a few genes with high fold-changes in LDM-positive plants, compared to LDM-negative and those with significantly higher or lower transcript abundance. After filtering the DEGs between LDM-positive and LDM-negative cultivars, 3305 genes were identified, as seen in Figure 1A. For 100 genes, no Pfam class [45] and, for a further 168, no EggNOG [46] description, could be assigned. Regarding the DEGs, a closer look reveals the 20 most diverged genes, which can be seen in Figure 1B,C. Half of the annotated genes are still uncharacterized, or their distinct function is unknown, according to the cluster of orthologous groups. Interestingly, the genes associated with metal transport and metal binding are differentially transcribed, as seen for g4372, g9694, and g8497. These functions are known to be responsible for catalyzing redox reactions in plants [47,48]. Examining DEGs further, g14432 is associated with the protein argonaute family and g1887 is a zinc finger-like protein, whereas g3464 is a thioredoxin/disulfide isomerase. These proteins regulate biological processes [49], as well as responses to abiotic stresses such as drought stress [50,51]. In general, these DEGs describe the effects on the primary metabolism and stress response of plants; however, they do not show any direct participation in tailoring secondary metabolites within the plants. CYPs, in particular, are iron-binding; however, a connection between the upregulation of metal-transporting proteins and CYPs cannot be drawn from this data. The biosynthesis of LDM is not artificially induced in one cultivar or silenced in the other. Thus, a specific and significant transcription of related terpene-tailoring genes cannot be observed. To elucidate these mechanisms, it is necessary to take a closer look into the DEGs of CYPs [28,52].

### 2.3. Terpene Tailoring through CYPs between Plant Cultivars

The identified 3305 DEGs can be further filtered into genes related to CYPs due to conserved domains and the corresponding CYP Pfam class. Here, the domain PF00067 was integrated into IPR001128. Both domains are indicators for sequences associated with the cytochrome p450 superfamily (IPR036396) [53]. This homology-based search allowed the identification of 70 putative sequences with different total lengths. Assuming a minimum size of 29 kDa for a CYP, 61 genes remain. From a statistical point of view, the average size of this pool amounts to a median of 1485 nucleotides, corresponding to the average size of a translated protein of 54.5 kDa. This is also reported in the literature, with an average plant CYP molecular mass between 45 and 62 kDa [54,55]. In regards to the identification of LDM-modifying enzymes, this subset is necessary to obtain a detailed overview into CYPs. These enzymes are known to play a huge part in terpene diversity in plants [56]. They are able to catalyze the hydroxylation of different backbones due to their substrate promiscuity [29,57]. Therefore, the transcript abundance of specific CYPs may reveal the mechanism behind LDM variances in this plant.

Out of all the 23,477 mapped genes, 221 CYPs were detected, whereas 61 showed differences in transcript abundance. In Figure 2, all identified CYPs are visualized in an unrooted phylogenetic tree. CYPs with high transcript abundance in LDM-positive cultivars are highlighted in green, whereas CYPs with low transcript abundance are represented in red.

To allocate the putative CYPs to their distinct family or subfamily, the Pfam-classified CYP sequences were subjected to a BLAST search using a custom CYP database [54]. The sequences were assigned to the same subfamily if the percent identity was above 55%, and to the same family if greater than 40%. Eight CYP clans were highlighted within the found enzymes, CLAN51, CLAN71, CLAN72, CLAN74, CLAN710, CLAN85, CLAN86, and CLAN 97. This highlights that the major classes 71 and 72 are found to be involved, primarily, in the terpene tailoring of different terpene classes [28]. For CYP71, a variety of monoterpene modifications are described [34,58,59,60,61]. In our setting, most DEGs were observed in this clan. The enzymes related to CYP72 are described as tailoring triterpenoids as saponins, characterized within plant defensive mechanisms against biotic stressors such as herbivores or microbes [38,62].

DEGs with high transcript abundance in LDM-positive samples were used to compare the genes between all sequenced cultivars. PP and SB were considered highly LDM-positive, whereas DK and GB were LDM-negative. GG and HG were in between and, therefore, were excluded in the initial DEG analysis. For the comparison of CYPs between the four previously mentioned samples and the two latter samples, the CYPs found in LDM-positive and LDM-negative samples were searched in GG and HG, and the normalized counts of all samples were compared. PP was chosen, due to its LDM profile, as a setpoint to compare the transcript abundance between all samples. In, the results of a comparative approach are visualized. The phylogenetic distance between the identified CYPs is shown in 3A. Three clusters can be differentiated, with the first seen in the upper part consisting of 4 genes (g25953, g25443, g578, g8489), the second in the middle (g3273, g10380, g27034, g27468, g27861), and the third cluster with 14 genes (g24222, g2313, g27787, g24257, g20804, g3860, g10700, g16684, g24219, g9390, g14070, g28342, g8554, g2205) at the bottom. In Section 3B, the fold-change between the cultivars is visualized; boxes marked with X were transcripts with no mapping results in the respective cultivar. The clusters do not share a similar transcript abundance pattern, nor do the genes that are closely related. However, investigating the recurring, fixed-length patterns inside the sequences led to the discovery of five motifs shared among all sequences. Figure 3C visualizes the motifs and their distribution in the sequence. The exact motif sequences are presented in Appendix A. A closer look also reveals distinct recurring, CYP-specific domains [63]. The conserved regions were reviewed extensively [38] and can be confirmed in this dataset. Starting with the proline-rich membrane hinge (motif 8), which is part of the membrane anchor, another conserved motif, which is important for the correct function of CYPs, is the site for oxygen binding and activation, A/G-G-X-E/D-T-T/S (motif 3). This is followed by the E-R-R triad and P(E)R(F) domain. Furthermore, the heme-binding site, with cysteine as the main ligand to the heme, C-X-G (motif 2), which is necessary for the typical redox reaction of CYPs [64], as well as the ERR triad (motif 6) and the (P(E)R(F)) sequence (motif 6), can be differentiated among the discovered 10 motifs.

Regarding the production of LDM, the genes g8554, g27861, g10700, and g24222 show an interesting pattern compared to the highly LDM-positive cultivar PP, which makes them candidates for further functional characterization to prove their LDM-producing potential.

The candidate genes were further investigated in terms of their putative function. The initial estimates, using sequence and structural homology, consider g2422 and g8554 to be involved in the hydroxylation of cinnamic acid, whereas g27861 and g10700 display unknown activity towards flavonoids, sterols, and ferruginol. This substrate promiscuity is known for CYPs, as they are able to catalyze different ligands [57,65], thus making functional characterization using prokaryotic, yeast, or plant expression systems indispensable to support claims on putative functions.

## 3. Materials and Methods

### 3.1. Plant Material

Cultivars of *Caryopteris × clandonensis*, DK, GB, GG, HG, SB, and PP, were acquired from a local nursery (Foerstner Pflanzen GmbH, Bietigheim-Bissingen). DK and GB were investigated to show a highly LDM-negative profile, whereas SB and PP show a highly LDM-positive profile. GG and HG showed a non-conclusive profile in between. After growing to maturity in the open in a warm, moderate climate zone, healthy leaves were sampled and snap-frozen in liquid nitrogen and stored at −80 °C until RNA preparation for RNA-Seq.

### 3.2. Genomic Resource

The reference genome of *Caryopteris × clandonensis* used in this study was obtained from NCBI SAMN32308290 (PP). The raw data were assembled as previously described [1] and subjected to further refinements. For further processing, the reference was cleared from possible contaminations, and scaled down from 783 contigs to 53 contigs using Metabat2 (v2.15) [42], keeping the genome completeness with 96.5% at a high level according to BUSCO (v5.3.2) [66] analysis (2326 BUSCO groups, lineage dataset: *Eudicotidae*). Gene model prediction was conducted using AUGUSTUS [67,68,69,70]. To detect repetitive sequences, such as tandem repeats or transposable elements, soft masking was employed using Red (v2018.09.10) [71].

### 3.3. RNA Preparation and Short Read Sequencing

High-quality RNA was extracted using the RNeasy Plant Mini Kit (Qiagen, Venlo, The Netherlands) according to the manufacturer’s protocol. To ensure RNA integrity, the Bioanalyzer RNA 6000 assay kit (Agilent, Santa Clara, CA, USA) was employed to yield an average RNA Integrity Number of 7.7. The library preparation was performed using the Illumina stranded mRNA prep kit with IDT for Illumina UD Indexes, Plate A. Corresponding adapter was the Illumina Nextera Adapter (CTGTCTCTTATACACATCT). Library preparation was performed according to the manufacturer’s protocol with a shortened fragmentation time from 8 min (protocol) to 2 min (this study). Sequencing was performed at the Helmholtz Munich (HMGU) by the Genomics Core Facility on a NovaSeq6000 SP (2 × 150 bp). For each sample, two lanes were loaded and an average of 22 Mio fragments were yielded. The corresponding lanes of each sample were concatenated tail-to-head (v8.25) [72]. The combined short reads were subjected to comprehensive quality control steps. Every step was analyzed with FastQC (v0.11.9) [73] and the necessity of another trimming step was evaluated. Sequences shorter than 20 bp minimum length and with a quality phred score beneath 20 were extracted from the paired-end read data. The Illumina Nextera Adapter was used to trim each read pair using Cutadapt (v.4.0) [74]. The first 10 bases were cut from the sequences, due to their sequence GC content, using Trimmomatic (v0.38) and headcrop parameter [75].

### 3.4. Mapping and Annotation of Aligned Reads

Refined short reads were mapped on the clean reference genome using STAR (v2.7.10b) [76], 140 bases were chosen as the length of the genomic sequence around annotated junctions. EggNOG (v2.1.5) [46,77] was employed to evaluate the function of the differentially expressed genes using Pfam, GO, and COG databases. MEME suite (v5.5.1) [78] was used for identification of motifs within sequences of interest. Visualizations were built in R. Except for STAR; all sequencing analyses were conducted using galaxy project [79]. Analysis was based on reference-based RNA-Seq data analysis [80,81]. The detection of CYPs was performed using a homology-based search, using the conserved domain PF00067, which was integrated to IPR001128. Both domains are indicators for a sequence association with the cytochrome p450 superfamily (IPR036396) [53]. CYP-family classification was performed using a BLAST search [82] and a custom database [83].

### 3.5. Evaluation of Differential Gene Expression between Aerial Plant Parts

Aligned transcripts were counted using FeatureCounts (v3.16) [84], normalized, and differentially investigated with DESeq2 (v1.34.0) [85,86,87]. An adjusted *p*-value below 0.05, and a fold-change greater than 2 and below 0.5, was used to determine the most differentially expressed genes in this dataset.

## 4. Conclusions

This study provides a basis for further CYP research in *Caryopteris × clandonensis*, especially regarding LDM. Furthermore, the reference genome was subjected to a cleaning step, resulting in a decrease from 782 scaffolds to 53 scaffolds. Six cultivars were subjected to an RNA analysis, which gradually neared the prediction of 4 possible LDM tailoring CYPs out of 24, which were differentially expressed, and showed high transcript abundance, compared to the other cultivars. Furthermore, the classification and phylogenetic analysis of all mapped CYPs were conducted and they showed a distinct clustering in CYP CLAN71 and 72. All essential and conserved motifs could be recognized within these sequences. However, experimentally focused research for functional characterization needs to be conducted in order to identify the exact predicted function of these enzymes. A further in silico step can include the prediction of docking and catalysis sites within a three-dimensional structural model, as well as through molecular dynamic techniques and free energy calculations [88,89].

In general, this approach can be used to detect further mechanisms and pathways in plants, which show valuable medicinal effects. The biotechnological production of artemisinin [90] and taxol [11] is a popular example of the possibilities in medicinal plant research. There are already several approaches used, which combine omics approaches to identify substances of interest [91,92,93].

## Figures and Tables

**Figure 1 plants-12-02305-f001:**
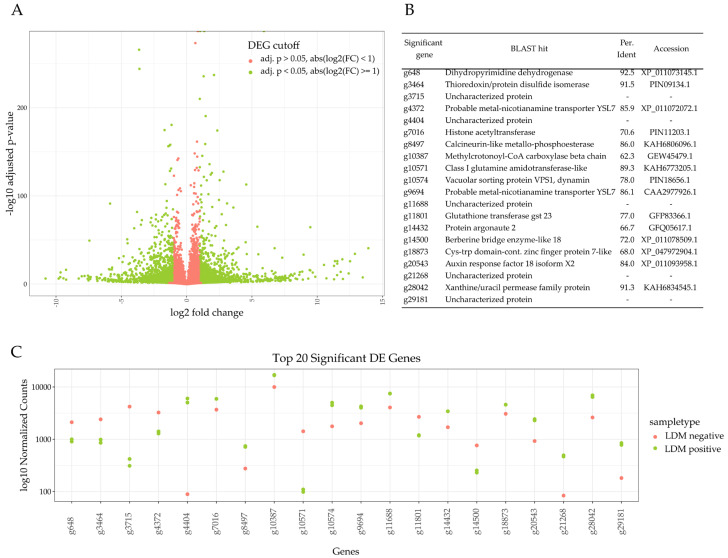
Differential expressed genes (DEGs) of *Caryopteris × clandonensis* cultivars highly producing limonene-derived molecules (LDM-positive) and cultivars which produce lower amounts of LDM (LDM-negative). Cultivars used for LDM-positive subset: Sunny Blue and Pink Perfection, and for LDM-negative subset Dark Knight and Grand Bleu. (**A**) The volcano plot of DEG was identified between the LDM-positive vs. LDM-negative plant cultivar subsets. Absolute log2 fold-change cutoff was set to 1 and an adjusted *p*-value of 0.05 was used to assign the DEGs; values fitting these parameters are highlighted in green and those which were disregarded during further analysis are shown in red. (**B**) Top 20 most significantly transcribed genes and their respective description, including BLAST search percentage identity and determined accession for the putative assignment. (**C**) log10 normalized counts of the top 20 significant DEG in this setup. Genes from LDM-positive samples are displayed in green, those corresponding to LDM-negative samples are highlighted in red.

**Figure 2 plants-12-02305-f002:**
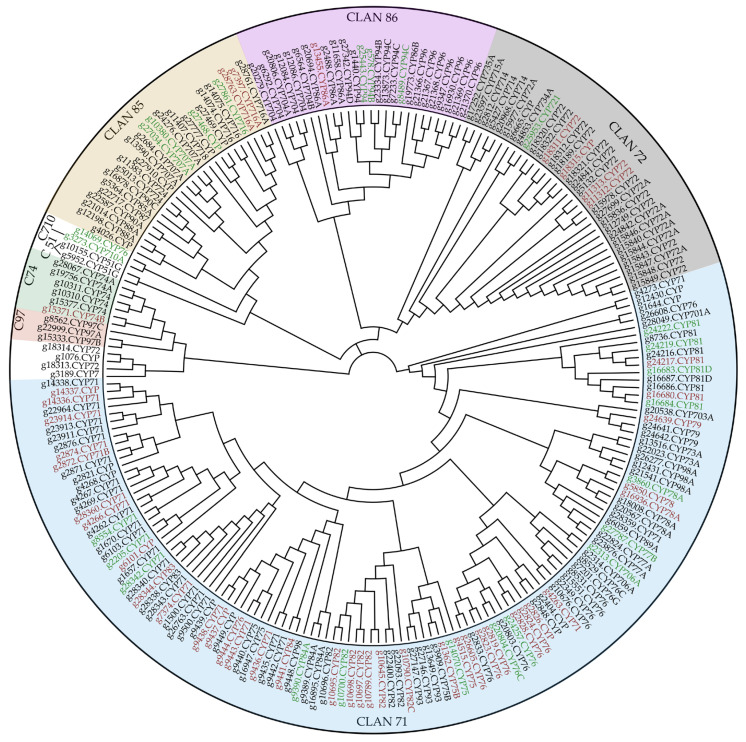
Phylogenetic tree of all transcribed cytochrome p450 (CYP) enzymes within the six investigated cultivars. Clan localization is highlighted on the outer ring. Differentially expressed genes (DEGs) were marked in green for a high transcript abundance in limonene-derived-molecules-positive cultivars and red for low transcript abundance, as seen in their fold-change differences. The tree was constructed using the following parameters: Global alignment with a Blosum62 cost matrix, Genetic distance model Jukes-Cantor, Neighbor-Joining and no outgroup was used, gap open penalty was set to 12, and gap extension penalty to 3 during pairwise alignments.

**Figure 3 plants-12-02305-f003:**
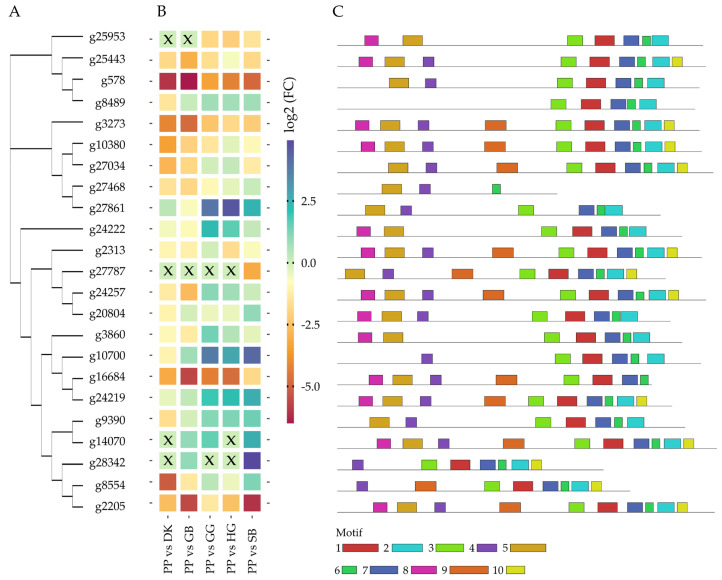
Analysis of differentially expressed cytochrome p450 enzymes (CYP) in different plant cultivars of *Caryopteris × clandonensis*, Dark Knight (DK), Grand Bleu (GB), Good as Gold (GG), Hint of Gold (HG), Sunny Blue (SB), and Pink Perfection (PP). (**A**) Phylogenetic analysis of CYP sequences with highly abundant transcripts regarding limonene-derived molecules (LDM) within the cultivars, using Neighbor-joining method. (**B**) Heatmap of normalized transcript counts between distinct cultivars. X represents enzymes with no transcripts in respective cultivars. The color palette displays genes with high transcripts abundance in red to light-yellow colors, high transcript abundance is depicted in light-green to blue (**C**) Identification of recurring, fixed-length patterns (motifs) identified in LDM-positive transcripts. Motifs 1 to 10 are illustrated as colored boxes, to distinguish the motifs between the different genes. Sequences can be found in Appendix A.

**Table 1 plants-12-02305-t001:** Statistics of short Illumina reads used for mapping on the reference genome (NCBI SAMN32308290 (Pink Perfection, PP)). A paired-end run was employed on a NovaSeq6000 SP (2 × 150 bp) for sequencing.

*Caryopteris × clandonensis* Cultivar		Raw Reads in Bases		Q20 in %	Q30 in %	Clean Reads in Bases		Q20 in %	Q30 in %	Totally Mapped in %	Uniquely Mapped in %
		Unique	Duplicate			Unique	Duplicate				
Dark Knight	R1	24,501,785	19,238,555	99.95	94.76	13,072,273	29,945,355	99.99	95.08	87.8	79.3
	R2	26,380,719	17,359,621	99.25	87.90	16,204,470	26,813,158	99.46	88.25		
Grand Bleu	R1	17,917,215	51,971,129	99.85	93.75	11,552,426	57,659,626	99.98	94.21	85.8	76.8
	R2	18,808,258	51,080,086	99.51	92.12	13,260,446	55,951,606	99.68	92.42		
Good as Gold	R1	22,797,327	27,074,692	99.60	94.52	13,160,322	31,359,084	99.95	95.08	86.7	75.4
	R2	25,142,438	24,729,581	99.35	89.41	16,112,061	28,407,345	99.54	89.76		
Hint of Gold	R1	20,547,645	20,953,229	99.89	94.67	15,165,071	33,935,373	99.98	95.06	86.4	77.2
	R2	23,044,700	18,456,174	99.38	88.40	18,084,814	31,015,630	99.56	88.81		
Sunny Blue	R1	20,535,582	25,022,034	99.96	94.96	13,908,152	27,181,140	99.99	95.35	87.0	80.5
	R2	22,771,085	22,786,531	99.39	88.30	16,573,745	24,515,547	99.56	88.60		
Pink Perfection	R1	25,751,312	28,610,858	99.96	94.23	12,295,625	26,046,846	99.99	94.60	87.7	82.0
	R2	29,512,685	24,849,485	99.40	90.42	14,649,539	23,692,932	99.58	90.70		

## Data Availability

Data are available in a publicly accessible repository. The refined genome data presented in this study are openly available at the National Center for Biotechnology Information (NCBI). BioSample accession number: Pink Perfection SAMN32308290.

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
