# Peer review of "Differential RNA-Seq Analysis Predicts Genes Related to Terpene Tailoring in Caryopteris × clandonensis"

_plants, 2023, doi:10.3390/plants12122305_

Round 1
Reviewer 1 Report
The authors claimed to improve the previously constructed genome assembly of Caryopteris x clandonensis by analysing the raw data and re-assembling it after removal of contaminating sequences, scaled down the assembly from 783 contigs to 53 contigs without compromising the assembly completeness. However, it is difficult to follow what authors have done at this point. In the methodology section, it is written that the raw data was assembled as previously described [1] and subjected to further refinements (Lines 245-246) and in the results and discussion section, it is written that the available sequence from Caryopteris x clandonensis PP [1] was used as a reference for mapping the short reads of all cultivars (Lines 113-114). Here the reader gets confused because it not clear whether the authors have used RNAseq reads generated in this study or the raw genomic sequences from [1] for mapping. Further, if the authors have refined the previous assembly, then why did they use previous reference as mentioned in line 113. What I understand from the methodology is that authors have first refined the available genome assembly, then aligned their RNAseq reads against this refined assembly. But in the results and discussion section, this is not clearly reflected.
The authors have generated RNAseq data of 6 different cultivars and have chosen 4 for DGE analysis of LDM because they showed contrasting LDM profile, but is not well stated in the manuscript in the results and discussion section, reflecting the poor organization of manuscript. Also what about the remaining two cultivars. Then after performing the DGE analysis, the authors did not bother to tell the readers about how many genes were upregulated or down-regulated in each comparison. By using four cultivars for DGE for LDM, at least 4 comparisons are possible between the contrasting LDM cultivars, but authors have shown only one volcano plot. I am not satisfied at this point regarding the presentation of results. As for the volcano plot shown in the manuscript, what does this correspond to? SB vs DK, SB vs GB, PP vs DK or PP vs GB. Have the authors used merged DGE results for LDM +ve and LDM-ve cultivars. This must be highlighted in manuscript.
Nowhere in the manuscript, the authors have mentioned in detail how they identified the CYPs and then classified them. There is only a feeble hint in lines 186 and 279. It not mentioned either how many they identified. Then suddenly, the authors jumped to line 184 where they write: In Figure 2 the sequences of all CYPs, which were mapped to the reference genome are visualized in an unrooted phylogenetic tree. How would reader know what is happening here.
The overall manuscript lack the novelty of the claims and poor representation of the claimed data. Further, neither methodology nor results organized properly. Even the results do not reflect the claims of the manuscript. Therefore, the current manuscript cannot be recommended publication.
Need significant improvement
Author Response
Dear reviewer,
please find enclosed the revised manuscript.We appreciate the time and effort you and each of the reviewers have dedicated to providing insightful feedback, to improve the quality of our manuscript.

Reviewer 2 Report
This manuscript reports the investigation on variations and additionally genes with high and low transcript abundancies in leaves of the plant, Caryopteris x clandonensis, aiming at terpene functionalization. This work predicted the cytochrome p450 enzymes, which have varied transcription pattern between cultivars of Caryopteris x clandonensis. It was found that the terpene variety between different plant cultivars can be guided by a molecular method and a quantitative bioinformatics method such as RNA-Seq analysis. Therefore, there is a reasonable explanation for terpenoid differences between among the cultivars.
The comments and suggestions are listed below.
1. The keyword “terpene tailoring” should be revised “terpene biosynthesis”. The word “terpene tailoring” is already in the title.
2. Keywords should include the words “chemical diversity” and “volatile compound”.
3. Any reasons why choosing the cultivars Dark Knight and Pink Perfection as the main subjects for this work.
4. Figure 2 shows hylogenetic tree of transcribed cytochrome p450 (CYP) enzymes, which have many CYPs found in the plant. How do you identify which one is LDM-positive cultivar or LDM-negative cultivar? Were these CYPs reported in plant only? Are they in microorganisms? In my opinion, the CYPs presented in this Figure should be those present in plants only.
5. “The raw data was assembled as previously described [1]”; please use “were”. The word “data” is plural.
6. The end of “Results and Discussion” needs discussion on the impact of this work. Limonene itself has been reported to have interesting biological activities, i.e. virucidal activity and being as a potential household disinfectant against virus (Natural Product Communications 2022, 17 (1), 1934578X211072713). Limonene-derived molecules from such terpene tailoring (genes) enzymes would produce a variety of limonene derivatives, which may have diverse biological activities. Would it be useful to apply this kind of work for other medicinal plants?
Minor English editing may be required.
Author Response

(The authors gave the same response as above.)

Reviewer 3 Report
In the submitted manuscript Ritz et al. showed that the identification of terpene variety between six different plant cultivars of Caryopteris x clandonensis can be pursued on a molecular level using a quantitative bioinformatics method, namely RNA-Seq analysis. Furthermore, the authors focused on terpene functionalization enzymes, namely cytochrome p450, to elucidate mechanisms behind the variations in modifications of limonene-derived molecules (LDM).
The subject of this manuscript is relevant to the field; however, there are a few comments that should be addressed before the final publication of the manuscript:
Introduction
The new Figure of Caryopteris x clandonensis would be very beneficial for the readers.
Lines 48-49: Known beneficial biological effects of terpenes, especially monoterpenes, should be briefly described. The biological effects of monoterpenes were comprehensively reviewed in recent articles.
References:
1. Furlan, V.; Bren, U. Helichrysum italicum: From Extraction, Distillation, and Encapsulation Techniques to Beneficial Health Effects. Foods 2023, 12, 802.
2. Lešnik, S., Furlan, V., and Bren, U. (2021) Rosemary (Rosmarinus officinalis L.): extraction techniques, analytical methods, and health-promoting biological effects. Phytochemistry Reviews, 1-56.
Section Results and Discussion
Subsection 2.2. Identification of DEG
Lines 158-162: The functions of characterized proteins in Figure 1B should be discussed.
Figure 1: The colors of DEG should be explained in the figure caption.
The obtained results indicate that the extracts of the hydrodistilled leaves possess a higher content of identified compounds than the non-hydrodistilled leaves. These findings need to be discussed in more detail in the corresponding subsections 2.2.1. to 2.2.5.
Subsection 2.3 Terpene tailoring through CYPs between plant cultivars
Line 212: The first cluster (Figure 3A) consists of four genes, not three (g25953, g25443, g578, and g8489).
Line 219-221: The motifs of typical recurring CYP-specific domains should be discussed in more detail.
Figure 3B and 3C: The color palette of distinct cultivars as well as motifs should be explained in the figure caption.
Section Conclusions
Lines 290-291: The names of four predicted CYP enzymes should be provided.
Further insights into molecular mechanisms of limonene-derived molecules from Caryopteris x clandonensis on various CYP proteins can be revealed through advanced molecular dynamics techniques and free-energy calculations as well as through inverse molecular docking.
References:
1. Pantiora, P.; Furlan, V.; Matiadis, D.; Mavroidi, B.; Perperopoulou, F.; Papageorgiou, A.C.; Sagnou, M.; Bren, U.; Pelecanou, M.; Labrou, N.E. Monocarbonyl Curcumin Analogues as Potent Inhibitors against Human Glutathione Transferase P1-1. Antioxidants 2023, 12, 63. https://doi.org/10.3390/antiox12010063
2. Kores, K.; Kolenc, Z.; Furlan, V.; Bren, U. Inverse Molecular Docking Elucidating the Anticarcinogenic Potential of the Hop Natural Product Xanthohumol and Its Metabolites. Foods 2022, 11, 1253. https://doi.org/10.3390/foods11091253
Line 167: The sentence "Volcano plot of DEG identified between and LDM-positive vs. LDM-negative." should be revised.
Author Response

(The authors gave the same response as above.)
